# Human Placental NADPH Oxidase Mediates sFlt-1 and PlGF Secretion in Early Pregnancy: Exploration of the TGF-β1/p38 MAPK Pathways

**DOI:** 10.3390/antiox10020281

**Published:** 2021-02-12

**Authors:** Isabelle Hernandez, Audrey Chissey, Jean Guibourdenche, Roger Atasoy, Xavier Coumoul, Thierry Fournier, Jean-Louis Beaudeux, Amal Zerrad-Saadi

**Affiliations:** 1Université de Paris, INSERM UMR-S 1139, 3PHM, F-75006 Paris, France; audrey.chissey@parisdescartes.fr (A.C.); jean.guibourdenche@parisdescartes.fr (J.G.); roger.atasoy@gmail.com (R.A.); thierry.fournier@parisdescartes.fr (T.F.); jean-louis.beaudeux@parisdescartes.fr (J.-L.B.); 2UF d’hormonologie Adulte de Cochin AP-HP, Hôpitaux Universitaires, F-75006 Paris, France; 3Université de Paris, INSERM UMR-S 1124, F-75006 Paris, France; xavier.coumoul@parisdescartes.fr

**Keywords:** pregnancy, first trimester, NADPH oxidase, p38 MAPK, SMAD2, sFlt-1, preeclampsia

## Abstract

Preeclampsia, a hypertensive disorder occurring during pregnancy, is characterized by excessive oxidative stress and trophoblast dysfunction with dysregulation of soluble Fms-like tyrosine kinase 1 (sFlt-1) and placental growth factor (PlGF) production. Nicotinamide Adenine Dinucleotide Phosphate (NADPH) oxidase (Nox) is the major source of placental superoxide in early pregnancy and its activation with the subsequent formation of superoxide has been demonstrated for various agents including Transforming Growth Factor beta-1 (TGF-β1), a well-known p38 MAPK pathway activator. However, the bridge between Nox and sFlt-1 remains unknown. The purpose of this study was to explore the possible signaling pathway of TGF-β1/Nox/p38 induced sFlt-1 production in human chorionic villi (CV). Methods: Human chorionic villi from first trimester placenta (7–9 Gestational Weeks (GW)) were treated with TGF-β1 or preincubated with p38 inhibitor, SB203580. For NADPH oxidase inhibition, CV were treated with diphenyleneiodonium (DPI). The protein levels of phospho-p38, p38, phospho-Mothers Against Decapentaplegic homolog 2 (SMAD2), and SMAD2 were detected by Western blot. The secretion of sFlt-1 and PlGF by chorionic villi were measured with Electrochemiluminescence Immunologic Assays, and NADPH oxidase activity was monitored by lucigenin method. Results: We demonstrate for the first time that NADPH oxidase is involved in sFlt-1 and PlGF secretion in first trimester chorionic villi. Indeed, the inhibition of Nox by DPI decreases sFlt-1, and increases PlGF secretions. We also demonstrate the involvement of p38 MAPK in sFlt-1 secretion and Nox activation as blocking the p38 MAPK phosphorylation decreases both sFlt-1 secretion and superoxide production. Nevertheless, TGF-β1-mediated p38 activation do not seem to be involved in regulation of the first trimester placental angiogenic balance and no crosstalk was found between SMAD2 and p38 MAPK pathways. Conclusions: Thus, the placental NADPH oxidase play a major role in mediating the signal transduction cascade of sFlt-1 production. Furthermore, we highlight for the first time the involvement of p38 activation in first trimester placental Nox activity.

## 1. Introduction

Preeclampsia (PE) is a hypertensive disorder that occurs during pregnancy and is one of the leading causes of death in pregnant women. It is the most common cause of premature labor. This vasospasm disease causes progressive damage in mothers’ tissues and multiorgan lesions, leading to complications including stroke, as well as hepatic and kidney failure. For the fetus, this may result in an intrauterine growth restriction (IUGR), which is recognized as an increased risk factor of long-term neurodevelopmental disorders and metabolic diseases in adulthood [1,2]. Besides aspirin, which can be beneficial before 16 gestational weeks (GW) for preventing preeclampsia, there is currently no curative treatment, and only childbirth and delivery of the placenta alleviate a mother’s symptoms. During pregnancy, the placenta supports the development and survival of the fetus. It is a hemochorionic organ, which allows the transfer of gas and nutrients to the fetus and also protects the fetus against toxics, viruses, and pathogenic agents. Its functional and structural unit in humans is the chorionic villi (CV), which is composed since 4 GW of a mesenchymal axis containing fetal vessels covered by a layer of mononucleated villous cytotrophoblast (VCT) which fuse to form the external polynucleated syncytiotrophoblast (ST) in direct contact with the maternal blood. The ST is the main endocrine tissue of the placenta which secretes in the maternal circulation steroid hormones (estradiol and progesterone), protein hormones, such as the human chorionic gonadotropin hormone (hCG), and angiogenic factors mainly of soluble Fms-like tyrosine kinase 1 (sFlt-1) and placental growth factor (PlGF).

Preeclampsia is a multifactorial and multisystemic disease of placental origin. Placental insufficiency plays a central role in the pathophysiology of the disease. A defective remodeling of the uterine vascularization during the first trimester leads to placental hypoperfusion and hypoxia-reoxygenation, inducing oxidative stress and trophoblast dysfunction, and provokes an increased imbalanced release of angiogenic trophoblastic factors PlGF and sFlt-1. This release in the maternal circulation results in both an excessive inflammatory response and endothelial dysfunction [3]. Although the cause of preeclampsia remains unclear, several clinical risk factors have been described and include nulliparity, maternal age (young or advanced maternal age), multiple gestation, obesity, African American race, assisted reproductive techniques, familial history of preeclampsia, and history of preeclampsia in a prior pregnancy [4,5].

Early intervention in pregnant women with high risk factors for preeclampsia or maternal screening during the first trimester of pregnancy can significantly improve clinical outcomes. Currently, in the management of PE, biological markers are used to improve the diagnosis and prognosis of this pathology; one of the most studied biomarkers is sFlt-1 [6,7,8], which is a soluble form of the vascular endothelial growth factor (VEGF) and PlGF receptor [9]. sFlt-1 binds to free VEGF and PlGF in the maternal circulation, thus reducing their bioavailability for their membrane receptor [10]. Recent studies have shown that in a high-risk population, PlGF in the first trimester is useful for predicting preeclampsia [11,12] and that since the second half of a pregnancy, a ratio of sFlt-1/PlGF lower than the threshold value of 38 has good negative predictive value, i.e., rules out the chance of short-term onset of PE [13].

In addition to sFlt-1 and PlGF, the levels of Transforming Growth Factor 1 (TGF-β1) and its soluble receptor (soluble endoglin) are also disturbed in sera of mothers with preeclampsia [14]. TGF-β1 is a growth factor involved in the control of cell growth, proliferation, differentiation, apoptosis, angiogenesis, immunoregulation, and epithelial-mesenchymal transition (EMT) [15,16]. One characteristic of EMT is cell migration and invasion. In placenta, TGF-β1 seems to be involved in extravillous cytotrophoblast invasion [17], activating through two serine/threonine transmembrane receptor kinases the SMAD2/7 pathways to initiate trophoblast growth and differentiation in first trimester chorionic villi explants [18]. Additional studies have shown that TGF-β1 is involved in reactive oxygen species (ROS) production through the activation of the NADPH oxidase (Nox) enzymes [19,20]. Excessive ROS production leads to oxidative stress, which is a characteristic of preeclampsia.

Interestingly, we previously demonstrate that ROS production in first trimester placenta occurs mainly through Nox activation [21]. Furthermore, Nox activity is increased in the early stages of pregnancy (before 10 GW) [21]. Nox is a prooxidant enzyme and its sole function is the production of superoxide anion (O_2_^−^) from molecular oxygen (O_2_). Several isoforms of Nox are described (Nox 1 to 5, Dual oxidase 1 (DUOX 1) and DUOX 2) and each of them possess catalytic subunits and requires a unique activation process through interaction and regulation of different subunits (p22phox, p47phox, p40phox, p67phox, Noxo1, and Noxa1). 

ROS has been demonstrated to activate the Mitogen Associated Protein Kinase (MAPK) pathways [22], and we previously demonstrated that in first trimester placenta, Nox activation is associated with p38 MAPK phosphorylation [21]. The p38 protein is a member of MAPK pathways, which are involved in many cellular functions in response to extracellular stimulus. p38 can be activated by a dual phosphorylation by Mitogen-activated protein Kinase Kinase 3 (MKK3) and MKK6 on its Thr-Gly-Tyr motif [23] and by MKK4 [24]. p38 MAPK is usually activated by environmental stresses (oxidative stress, UV radiation, osmotic shock, pro inflammatory cytokines, etc.) or growth factor such as TGF-β1 [25,26,27]. When activated, p38 is classically translocated to the nucleus to phosphorylate several transcription factors. Moreover, it has been shown that p38 MAPK is involved in the placental angiogenic balance in an extravillous cytotrophoblast experimental model (HTR8/SVneo). Liu and al. previously showed that inhibition of p38 decreases sFlt-1 secretion [28]. Furthermore, p38 activation is increased in preterm preeclampsia placenta, a phosphorylation which is stronger in the case of Hemolysis, Elevated Liver enzymes and Low Platelets (HELLP) syndrome, underlining the sensibility of p38 pathways to hypoxic/ischemic disorder and its possible involvement in preeclampsia pathophysiology [29].

In our study, we aim to investigate the complex interplay between the TGF- β1 and p38 MAPK pathways, as well as the regulation of NADPH oxidase and the secretion of sFlt-1/PlGF, two critical factors for the development of preeclampsia.

## 2. Materials and Methods

### 2.1. Ethical Statement

First trimester placental tissues were obtained from the patients from Cochin Port-Royal maternity units, after receiving their written informed consent. The protocol was approved by the institutional review board “Comité de Protection des Personnes 2015-mai-13909”.

### 2.2. Materials

Hank’s Balanced Salt Solution (HBSS, reference: 14175-053) and Phosphate Buffer Saline (PBS, reference: 14190-094) were from Thermo Fisher Scientific (Waltham, MA, USA), as were the protein assay kit Micro Bicinchoninic Acid (BCA)™ Assay Kit (reference: 23235), Triton X-100 (reference: 11891445) and Tissue-Protein Extraction Reagent T-PER™ (reference: 78510). Protease inhibitor cocktail (reference: 539131) and phosphatase inhibitor cocktail (reference: 524629) were from VWR (Radnor, PA, USA). Recombinant human TGF-β1 protein (reference: ab50036) was purchased from Abcam (Cambridge, UK) and SB203580 (reference: 559389) was purchased from Merck Millipore (Merck KGaA, Darmstadt, Germany). Diphenyleneiodonium (DPI, reference: D2926) was from Sigma Aldrich (Saint-Louis, MO, USA).

For NADPH oxidase activity assay, potassium dihydrogen phosphate (KH2PO4, reference: 1.05104) was from Merck Millipore (Burlington, MA, USA), glycol ether diamine tetraacetic acid (EGTA, reference: E-4378) was from Sigma Aldrich (Saint-Louis, MO, USA) and sucrose (reference: 35579) was from SERVA Electrophoresis GmbH (Heidelberg, Germany). Lucigenin (reference: 14872) was from Cayman (Ann Arbor, MI, USA) and nicotinamide adenine dinucleotide phosphate H (NADPH, reference: N7505) was from Sigma Aldrich (Saint-Louis, MO, USA).

For immunolocalization studies, agarose (reference: 444153H) was from VWR (Radnor, PA, USA) and paraformaldehyde (PFA, reference: 15710) was from Euromedex (Souffelweyersheim, Bas-Rhin, France). Human Immunoglobulin G (IgG) (ref: 009-000-003) and goat serum (reference: 005-000-121) were from Jackson ImmunoRechearch (Ely, UK). Alexa Fluor 488, 546 and DyLight Fluor-conjugated antibody (600 or 800 conjugate) were purchased from Life Technologies (Carlsbad, CA, USA). DAPI (reference: D9542) was purchased from Sigma Aldrich (St. Louis, MO, USA) and Bovine serum albumin IgG-free (BSA, reference: 011-000-162) was from Interchim (San Diego, CA, USA). Monoclonal mouse anti-human cytokeratin 7 (clone OV-TL 12/30, reference: M7018) was from Dako (Agilent Technologies (Santa Clara, CA, USA)). Anti-phospho-p38 MAPK (Thr180/Tyr182) XP^®^ rabbit monoclonal antibody (reference: 4511), anti-p38 MAPK XP^®^ rabbit monoclonal antibody (reference: 8690), anti-phospho-SMAD2 (Ser465/467) rabbit monoclonal antibody (reference: 138D4) and anti-SMAD2 XP^®^ rabbit monoclonal antibody (reference: D43B4) were purchased from Cell Signaling Technology (Danvers, MA, USA). Anti-vinculin mouse monoclonal antibody (reference: V9131) was from Sigma Aldrich (St. Louis, MO, USA). sFlt-1 assays were performed using Electrochemiluminescence Immunologic assays (reference: 05109523, Roche Diagnostics, Switzerland) on a Cobas e analyzer.

### 2.3. Placenta Retrieval, Chorionic Villi Storage and Culture

All placentas were obtained from singleton pregnancy after voluntary elective terminations of pregnancy in agreement with the French law. No placenta were retrieved from assisted reproductive technology (ART) procedure. These abortion procedures were conducted in cases of presumably normal fetuses. In the case of miscarriages or fetal abnormality diagnosis prior to the chirurgical intervention, placentas were removed from the study. No patient was addressed in a context of morphological anomaly or fetal genetic disease.

Chorionic villi were collected from first trimester placenta (7–9 GWs) and washed within warm (37 °C) HBSS 1X, then chorionic villi were dissected free of membranes, quickly frozen and stored at −80 °C, or cultured in 24 well culture plate in a volume of 2 mL of medium (Dulbecco’s Modified Eagle Medium (DMEM) without phenol red, glucose 1 g/L, 10% decomplemented fetal calf serum, 1% glutamine, 1% penicillin-streptomycin) for 96 h, in 20% O_2_, 5% CO_2_, and 37 °C atmosphere, with or without the addition of treatment, and snap-frozen prior to −80 °C storage. 

### 2.4. TGF-β1 and SB203580 Treatments

After 24 h of culture, chorionic villi were either treated with 10 ng/mL TGF-β1 [18] for 48 h or preincubated with 10 µM SB203580 [30] for 1 h followed with 48 h of 10 ng/mL TGF-β1 treatment. For NADPH oxidase inhibition, chorionic villi were treated after 24 h of culture with 2,5 µM diphenyleneiodonium (DPI) for 1 h. Supernatants were retrieved every day to monitor sFlt-1 secretion.

### 2.5. Immunostaining

Immunohistochemistry was performed on human 7–9 GW chorionic villi, as previously described [21]. Briefly, fresh samples were fixed in paraformaldehyde (4%, for 4 h at 4 °C, and then 1% PFA at 4 °C overnight), then embedded in 4% (w/v) agarose. Sections (80–120 µm) were permeabilized (0.5% Triton X-100) and blocked in 3% BSA IgG-free, 5% goat serum, human IgG (final concentration: 12.5 µg/mL), and 0.01% Triton X100. Then, sections were incubated with primary antibodies (anti-total p38 (0.25 µg/mL) and anti-cytokeratin 7 (1.3 µg/mL)) followed by the appropriate fluorochrome-conjugated secondary antibody (1:500 Alexa Fluor 488 or 546). Nuclei were stained with DAPI (0.2 µg/mL).

Acquisitions were performed with a confocal microscope Leica^®^ SP2 (Leica Microsystems GmbH, Wetzlar, Germany) and analyzed with ImageJ software (Bethesda, MD, USA).

### 2.6. Western Blot Analysis

Protein extraction from first trimester villi was performed as previously described [21]. Briefly, villi were homogenized using commercial extracting buffer (T-PER™ containing 0.01%, protease inhibitor cocktail and 0.01% phosphatase inhibitor cocktail) using Ultra-Turrax^®^, then centrifuged 10 min at 11,000 g at 4 °C. Supernatants were collected and then stored at −80 °C.

Protein concentrations were assayed using a Micro BCA™ Assay Kit following the manufacturer’s instructions and plate read using a spectrophotometer Enspire^®^ (Perkin Elmer, Waltham, MA, USA). 

Samples were boiled for 5 min and protein extracts were resolved by 10% SDS-PAGE. Then immunoblots were incubated with primary antibodies anti-total p38 (1:1000), anti-phospho p38 (1:1000), anti-total SMAD2 (1:1000), anti-phospho SMAD2 (1:1000), or anti-vinculin (1:1000) followed by appropriate DyLight Fluor-conjugated secondary antibody (680 or 800 conjugate, dilution 1:20 000). Blots were revealed by using an Odyssey infrared imaging system, and then analyzed with Odyssey application software v3.0 (Li-Cor Bioscience, Lincoln, NE, USA). 

### 2.7. Measurement of NADPH Oxidase Activity

NADPH oxidase activity assay was adapted from Raijmakers et al. [31]. Frozen villi were thawed in a cold buffer solution, pH 7.4, containing 50 mM KH2PO4, 1 mM EGTA, 150 mM sucrose and 0.1% protease inhibitor cocktail, and then homogenized with an Ultra-Turrax^®^ (GmbH and Co. KG, Esslingen, Germany). Each well was filled with the sample homogenate (18 µL), 5 µM lucigenin and 400 µM NADPH in a total volume of 90 µL reaction buffer. Following 10 min of dark adaptation, luminescence (arbitrary light units (ALU)) was monitored at 37 °C for 30 min using a microplate reader Enspire^®^ 2300 Multilabel Reader^®^ (PerkinElmer, Waltham, MA, USA). Assays were performed in triplicate. The protein concentration was determined using the Micro BCA™ Assay Kit. For each assay, the area under the curve (AUC) was calculated and considered as the superoxide production.

### 2.8. Measurement of sFlt-1 Secretion

Free sFlt-1 levels were assessed in chorionic villi supernatants using the ElectroChemiLuminescence Immunologic Assay (ref: 05109523), by a Cobas 8000 analyzer (Roche Diagnostics, Switzerland) according to International Organization for Standardization ISO 15,189 and the manufacturer’s instructions. The values of sFlt-1 levels were normalized on the weight of chorionic villi explants. 

### 2.9. Statistical Analysis 

All data were analyzed with GraphPad Prism software (La Jolla, CA, USA). For more than two groups comparison, RM one-way ANOVA statistical analysis followed by Tukey’s test when significance was applied for parametric data. Friedman test followed by Dunn’s post-hoc test when significance was applied for non-parametric data. Paired t-test for parametric or Wilcoxon test for non-parametric data were applied for comparison of two groups. A *p*-value lower than 0.05 was considered to be statistically significant, with *p* < 0.05 and *p* < 0.01 represented as * or **, respectively. Graphical representations show experimental results with median ± interquartile range (non-parametric analysis) or mean ± SEM (parametric analysis).

## 3. Results

### 3.1. Studied Population

For this study, a total of 31 women were included. The mean age of the patients was 29.1 years (SD: +/− 5.6 years). Among the patients, 41.93% were smokers at the time of the recruitment. The mean gestational age was 7 GWs (SD: +/− 1.3 weeks).

### 3.2. NADPH Oxidase Activity Modulation Using TGF-β1

In this study, we first assayed the kinetics of Nox activity in 7–9 GW chorionic villi treated with 10 ng/mL TGF-β1 for 30 min to 4 h. Superoxide production (representative of Nox activity) was measured using the lucigenin method, but no significant variation of Nox activity was observed between the different times of treatment (Figure 1A, Friedman test, *p* = 0.1338 and *n* = 5 for each group, control: untreated chorionic villi). To evaluate the effect of a more “chronic” exposure to TGF-β1, mimicking the preeclampsia process, and to decipher whether p38 MAPK may be involved in Nox activation, we monitored the enzyme activity in first trimester chorionic villi treated with 10 ng/mL TGF-β1 for 48 h with or without pretreatment with 10 µM SB203580 (a p38 MAPK inhibitor referred as SB) for 1 h. The production of superoxide by Nox under TGF-β1 treatment displayed no significant variation as compared with the vehicle group. However, the pretreatment with SB203580 prior to TGF-β1 activation led to a significant 1.6-fold decrease in superoxide production as compared with the vehicle group (Friedman test, *p* = 0.0125). No statistically difference was found between the TGF-β1 and SB + TGF-β1 groups, although a 1.2-fold decrease was observed when villi were pre-treated with p38 MAPK inhibitor (Figure 1B, *n* = 13, vehicle: DiMethyl SulfOxide (DMSO)).

p38 MAPK usually activated by environmental stresses and involved in the placental angiogenic balance, regulates sFlt-1 secretion. We investigated its expression profile in our system (Figure 2) and subsequently studied the crosstalk between its activation and the TGF-β1 signaling pathway (Figure 3 and Figure 4).

### 3.3. p38 Protein Is Mostly Expressed in Villous Cytotrophoblast

The confocal microscopy histological examination of chorionic villi tissue showed that total p38 was mostly localized in villous cytotrophoblast (white arrows, Figure 2A–C). Its expression was drastically decreased in the syncytiotrophoblast and mesenchymal axis (Figure 2A–C).

### 3.4. TGF-β1-Mediated p38 MAPK Activation Can Be Prompt and Long-Term Phenomenon

In parallel with the Nox activity, we studied the TGF-β1-mediated p38 MAPK activation. The phosphorylated p38/total p38 protein ratio was calculated and normalized to vinculin expression in total lysate of treated chorionic villi. The phosphorylation state of p38 protein in 7–9 GW chorionic villi after different time exposure to 10 ng/mL of TGF-β1 was assayed. Phosphorylation was monitored for 0, 0.5, 1, 2, 4, and 48 h; the protein expression of phosphorylated over non-phosphorylated p38 was observed after 30 min of exposure. The ratio was multiplied by 1.8-fold at 30 min under TGF-β1 exposure and decreased thereafter at levels comparable to the control (T0) (Figure 3A, one-way ANOVA, *p* = 0.0038, *n* = 7, control: untreated chorionic villi). Pictures of the whole Western blot membranes are available in the Appendix A. When chorionic villi were treated with 10 ng/mL TGF-β1 for 48 h, our results displayed a more than 2-fold increase in the phosphorylated/total p38 protein ratio as compared with the vehicle control group. Furthermore, this effect was abolished when the villi were preincubated with the p38 MAPK inhibitor SB203580 10 µM for 1 h prior to 48 h of TGF-β1 treatment. We observed about a 1.8-fold significant decrease in the phosphorylated p38/total p38 protein ratio as compared with the TGF-β1 treated group (Figure 3B, RM one-way ANOVA, *p* = 0.0041, *n* = 11, vehicle: DMSO). No difference was found between the SB + TGF-β1 and the control group, as illustrated in Figure 3B. Pictures of the whole Western blot membranes are available in the Appendix A.

### 3.5. TGF-β1 Canonical Pathway Study: Involvement of SMAD2

As TGF-β1 modulates p38 MAPK in our model, we further investigated one of the second messengers involved in the TGF-β1 canonical pathway to understand the underlining mechanism. The kinetic of SMAD2 phosphorylation (from 0.5 to 4 h) after treatment with TGF-β1 displayed a progressive increase, reaching 3.8-fold after 4 h treatment (Figure 4A, Friedman test, *p* = 0.0376, *n* = 3 independent experiments, control: untreated chorionic villi). Picture of the whole Western blot membranes are available in the Appendix A. To examine if any crosstalk between both p38 and TGF-β1 signaling pathways could occur, chorionic villi were treated with recombinant TGF-β1 with or without SB203580 (the p38 inhibitor). SMAD2 phosphorylation induced by TGF-β1 treatment for 48 h in 7–9 GW chorionic villi increased the phosphorylated SMAD2/total SMAD2 ratio by 1.5-fold (Figure 4B, RM-one way ANOVA, *p* = 0.0101, *n* = 7). However, the pretreatment with p38 MAPK inhibitor SB203580 did not affect the SMAD2 phosphorylation, indicating that p38 may be independently involved in TGF-β1/SMAD2-mediated sFlt-1 secretion (Figure 4B, *n* = 7, vehicle: DMSO). Pictures of the whole Western blot membranes are available in the Appendix A.

### 3.6. Implication of p38 MAPK Pathway and NADPH Oxidase on sFlt-1 Secretion and Angiogenic Balance

The sFlt-1 secretion was measured in first trimester chorionic villi supernatant cultured for 48 h with 10 ng/mL TGF-β1 or preincubated 1 h with 10 µM SB203580 followed with 48 h of 10 ng/mL TGF-β1. The results showed that the sFlt-1 increased by 1.2-fold when the chorionic villi were treated with TGF-β1 as compared with the vehicle group (Figure 5A, *p* = 0.1952, *n* = 7, vehicle: DMSO). When the villi were preincubated with the p38 inhibitor SB203580, the sFlt-1 secretion decreased by 1.6-fold as compared with the TGF-β1 group, but these results were not statistically significant (Figure 5A, paired t-test, *p* = 0.0675, *n* = 7 independent experiments). Interestingly, pre-incubation with p38 MAPK inhibitor significantly decreased the sFlt-1 secretion in chorionic villi supernatant as compared with the vehicle group (Figure 5A, paired t-test, *p* = 0.0308, *n* = 7, vehicle: DMSO), demonstrating the involvement of p38 MAPK in sFlt-1 secretion in our model. When first trimester chorionic villi were exposed to the Nox inhibitor DPI, the sFlt-1 secretion significantly decreases by 1.6-fold (mean +/− SD, 9949.25 +/− 3419.45 pg/mL versus 5569.00 +/− 2800.05 pg/mL, *p* = 0.0024), highlighting the involvement of Nox on sFlt-1 secretion (Figure 5B). The 2.5 µM DPI treatment on first trimester chorionic villi led to a significant increase by 3.3-fold in PlGF secretion (mean +/− SD, 39.96 +/− 12.89 pg/mL versus 93.89 +/− 51.86 pg/mL, *p* = 0.0218) and therefore to a significant decrease by 3.2-fold in the sFlt1/PlGF ratio when Nox was inhibited (Figure 5B, *p* = 0.0078, *n* = 8, vehicle: DMSO). 

## 4. Discussion

In this work, we showed for the first time that the sFlt-1 and PlGF secretion in first trimester chorionic villi are regulated by Nox activity, resulting in the modulation of the angiogenic balance in first trimester chorionic villi.

As TGF-β1 is increased in serum of preeclamptic women [14] and p38 is involved in extravillous cytotrophoblast sFlt-1 secretion [28], we initially hypothesized that p38-mediated sFlt-1 secretion in first trimester chorionic villi might be regulated by circulating TGF-β1 through the activation of placental Nox, which in turns activate the p38 MAPK cascade with ROS production.

In preeclamptic placenta, the Nox2 subunits are overexpressed [32] but its involvement in preeclampsia pathophysiology remains unclear. Raijmakers et al. showed an increased total superoxide production in placentas from early onset preeclampsia (<34 GW) as compared with late onset preeclampsia but the levels of superoxide remained unchanged between term and preeclamptic placentas [33], which clearly suggests that Nox may be involved in early onset preeclampsia. In our model, we observed a decrease in sFlt-1 secretion and an increase in PlGF secretion in presence of the Nox inhibitor DPI, a flavoprotein inhibitor of all Nox isoforms, which supports the relevance of our initial statement. Furthermore, our results suggest a crosstalk between the Nox and p38 MAPK pathway but the mechanism of action remains unclear. Nox activity seems to be under the influence of p38 MAPK activation as the blockade of its phosphorylation inhibited Nox activity and treatment of chorionic villi with DPI did not influence p38 MAPK phosphorylation (data not shown). In our model and experimental design, there is no significant effect of TGF-β1 on Nox activity. To explain this observation, the expression of both catalytic and cytosolic subunits of Nox isoform remains to be tested. TGF-β1 seems to have opposite effects depending on the cell type. In microglia cells, TGF-β1 prevents lipopolysaccharide (LPS)-induced NADPH oxidase subunit p47phox translocation from the cytosol to the membrane through p47phox-Ser345 phosphorylation inhibition [34]. However, in vascular smooth muscle cells (VSMC), TGF-β1 treatment triggers the Nox activity by increasing the expression of membrane cofactor, the p22phox subunit [35]. In human fetal lung fibroblast, TGF-β1 treatment increases H_2_O_2_ production and this effect can be reversed in the presence of the Nox inhibitor, DPI [19]. These opposite effects appear to depend on the Nox isoforms expressed in the cells. In addition, the level of H_2_O_2_ produced by human pulmonary fibroblasts begins to increase significantly after 4 h of exposure to TGF-β1 and reaches a peak at 16 h, to become undetectable at 48 h. [19]. Therefore, the possibility that similar kinetics occur in our model must be considered, and we may have missed the window of chorionic villi superoxide production.

In early (7–9 GW) first trimester chorionic villi, our results showed that TGF-β1 acts as a p38 MAPK pathway activator. The time exposure study revealed that an “acute-like” stress occurs after a 30 min exposure and a second activation is distinguishable after 48 h of treatment. In our study, we focused on the impact of the p38 MAPK activation by TGF-β1 for 48 h of treatment, corresponding to a more “chronic-like” scheme that may occurs during early onset preeclampsia as circulating TGF-β1 levels are increased [14,36]. TGF-β1 is a well-known p38 MAPK pathway activator [37]. Szabo et al. showed that activation of the p38 MAPK pathways is observed in placenta from women with preterm preeclampsia, and even higher when preeclampsia is complicated with HELLP syndrome. They also showed that hypoxia activates the p38 pathway. In addition, in Bewo cells, the inhibition of p38 pathway has the greatest impact on decreased Fms-Related Receptor Tyrosine Kinase 1 (FLT1) gene expression as compared with other MAPK pathways [29]. Interestingly, our results showed a decrease in sFlt-1 secretion when chorionic villi were treated with p38 MAPK inhibitor, but no impact of TGF-β1-mediated p38 activation was observed. This result can be explained by the complexity and the many different signaling cascades able to increase p38 MAPK phosphorylation. Despite evidence of a p38 MAPK involvement in sFlt-1 secretion, we do not exclude that TGF-β1-mediated p38 activation in our model, may not be the main pathway involved in the modulation of the angiogenic balance. Nevertheless, the tendency to decrease sFlt-1 secretion associated with a decrease in Nox activity when villi are treated with SB203580 strongly suggests its involvement in the angiogenic balance through an indirect mechanism possibly involving Nox. As the sFlt-1/PlGF ratio is a prognostic tool in PE in the second half of pregnancy [6], the impact of TGF-β1 mediated p38 MAPK activation on PlGF secretion remains to be tested in our model, to pursue the exploration of the impact of the TGF-β1/p38 pathway in PE pathogenesis. In our study, Nox activity inhibition led to an increase in PlGF secretion. Nox4 has previously been described as involved in PlGF secretion in a human coronary artery endothelial cells exposed to fluid shear-stress, as Nox inhibition with DPI attenuated the effects of shear-stress on PlGF protein expression [38]. As we demonstrated in first trimester chorionic villi, Nox inhibition using DPI also leads to PlGF secretion modulation, although the observations are opposite. The understanding of PlGF secretion regulation in early pregnancy remains essential in preeclampsia prevention, as it has been identified as an early predictive factor of the onset of PE [11]. As placental Nox isoforms still need to be clearly identified, the underlining mechanism remains to be explored to understand these opposite effects of Nox activation on PlGF induction.

To improve our understanding of the pathways involved, we further studied one second messenger of the TGF-β1 pathway. TGF-β1 fixation to its receptor Transforming Growth Factor Receptor II (TβRII) leads to the recruitment of Transforming Growth Factor Receptor I (TβRI) receptor (also named ALK5), and then activation of receptor regulated Smads (R-SMADs) proteins SMAD2/3. These proteins heterodimerize and bind to their partner named co-SMAD SMAD4 and are translocated together into the nucleus, playing a regulatory role on target genes. In first trimester chorionic villi, Xus’ group demonstrated that SMAD2 phosphorylation was higher in 6–7 GW chorionic villi as compared with later periods of pregnancy [18]. SMAD2 is located within the villous cytotrophoblast and syncytiotrophoblast nuclei [18]. Phosphorylated SMAD2 and SMAD7 expression are also increased in early onset preeclampsia as compared with an age matching control [18]. TGF-β1 level in maternal serum is increased in preeclampsia; the phagocytosis by endothelial cells of necrotic trophoblast released in maternal circulation leads to TGF-β1 secretion, which increases interleukin 6 (IL-6) expression, and then IL-6 induces activation of endothelial cells [39]. In our model, we also provide evidence of enhanced SMAD2 phosphorylation after TGF-β1-mediated p38 MAPK activation. SMAD2 is mainly described as a TGF-β1 target. Interestingly, we did not observe any crosstalk between p38 and SMAD2 pathways, suggesting that they may act independently in the regulation of Nox-mediated angiogenic balance modulation. To our knowledge, no direct involvement of SMAD2 on PlGF secretion has been described, but recent data have demonstrated the involvement of TGF-β1 and SMAD1/5 in prohaptoglobin-induced PlGF secretion in human umbilical vein endothelial cells [40]. In first trimester placenta, a study of TGF-β1 effectors revealed the presence of SMAD2/4 and 7 in villous cytotrophoblast and SMAD3 in extravillous cytotrophoblast [41]; suggesting that PlGF modulation in our model is independent of SMAD2 pathway.

To this day, there is a lack of data to demonstrate a link between TGF-β1 pathway, p38 MAPK activation and sFlt-1 secretion in first trimester chorionic villi. Hence, exploration of the pathways involved in sFlt-1 secretion and other angiogenic factors in first trimester placenta model may be more complex and may involve other pathways, such as Hypoxia-Inducible Factor 1 alpha (HIF-1α) [42]. Other increased biological markers may be regulated during first trimester environmental transition, such as soluble endoglin, which is increased in maternal sera in the case of preeclampsia [43]. The oxygen tension also plays a role in soluble endoglin (sEndo) secretion through TGF-β3 signaling [44] and emphasizes the complex interplay between several actors involved in angiogenic balance and oxygen tension during the first trimester. Poor trophoblastic invasion is observed in preeclamptic placentas and may be influenced by extravillous cytotrophoblast autophagy mediated by soluble endoglin [45].

In our model, we initially hypothesized that TGF-β1 mediates sFlt-1 secretion and involves p38 MAPK activation. Studying the canonical TGF-β1 pathway, we show that SMAD2 protein is phosphorylated 4 h after exposure to TGF-β1. Nevertheless, p38 MAPK inhibitor SB203580 action prior to TGF-β1 treatment did not lead to a decrease in SMAD2 phosphorylation. Thus, these results show in our model that SMAD2 activation is independent of p38 activation. The involvement of p38 MAPK in SMAD2 phosphorylation has been previously described in VSMC model [46] using another p38 MAPK inhibitor, SB202190. The study of the phosphorylation site offers a hypothesis to explain those results. There are two phosphorylation residues in SMAD2, localized in the C-terminal region or in the linker region [47]. C-terminal sites are phosphorylated during canonical activation of the TGF-β1 pathway by upstream proteins as ALK5, but linker sites are phosphorylated in the non-canonical pathway, for example by MAPK [41]. Nonetheless, p38 does not bind to the SMAD2 linker region, but SMAD3 [47]. Hence, the involvement of SMAD2 in sFlt-1 secretion cannot be excluded. In the same way, other upstream proteins may activate the p38 MAPK pathway after TGF-β1 exposure. For example, ALK5 interacts with TNF Receptor Associated Factor 6 (TRAF6), which recruits Transforming Growth Factor beta-Activated Kinase 1 (TAK1) and TAK1-binding protein 1 (TAB1) proteins. This protein complex is, in turn, able to phosphorylate p38 [48,49,50]. In addition, ALK5 expression and SMAD2 activation has been shown to be increased in early onset preeclampsia, but not in late onset preeclampsia [18]. Furthermore, ALK-5 has been demonstrated to upregulate PlGF gene expression in human umbilical vein endothelial cells [51], which remains to be tested in our model.

In this work, we emphasize that sFlt-1 secretion may be under the control of several independent activator pathways, including TGF-β1-mediated p38 MAPK activation and NADPH oxidase ROS production in the first trimester chorionic villi. However, p38 MAPK phosphorylation seems to be independent of SMAD2 activation (Figure 6). We also demonstrate for the first time that first trimester placental Nox activity can be under control of p38 MAPK phosphorylation, and that the p38 protein is mostly located in villous cytotrophoblast.

## 5. Conclusions

This study demonstrates for the first time the involvement of NADPH oxidase in sFlt-1 and PlGF secretions during the early stages of pregnancy (before 9 GW). It also shows the involvement of the p38 MAPK pathway in sFlt-1 secretion and, for the first time, in the activation of placental Nox. However, TGF-β1-mediated p38 activation does not modulate the SMAD2 pathway and does not seem to be involved in the regulation of the first trimester placental angiogenic balance. Hence, placental Nox plays a major role in in mediating the signal transduction cascade of sFlt-1 production, underlining its potential role in the pathogenesis of early onset preeclampsia. Furthermore, we highlighted for the first time the involvement of p38 activation in the first trimester placental Nox activity.

## Figures and Tables

**Figure 1 antioxidants-10-00281-f001:**
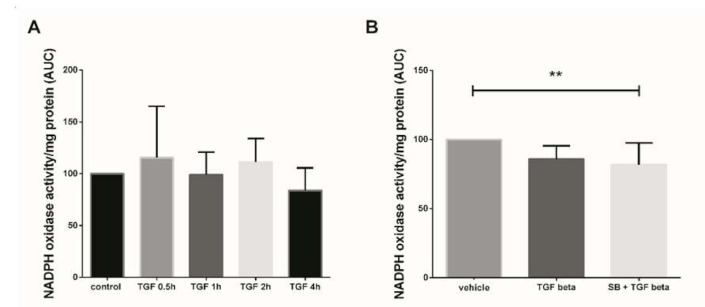
Effects of transforming growth factor 1 (TGF-β1) treatment on NADPH oxidase (Nox) activity. (**A**) Kinetics of Nox activity in first trimester chorionic villi exposed to 10 ng/mL TGF-β1. Superoxide production was monitored for 30 min. Results are expressed as area under curve and normalized on protein levels. Data are represented as median +/− interquartile range, *n* = 5 different placentas. Friedman test was applied; (**B**) Chorionic villi were treated with TGF-β1 (10 ng/mL) for 48 h with or without pretreatment with SB203580 (10 µM) for 1 h. Treatment with p38 MAPK inhibitor significantly decreases Nox activity. Data are represented as area under curve normalized on protein levels and expressed as a percentage of the vehicle group. Mean +/− SEM, ** *p* < 0.01, *n* = 13 different placentas. Friedman test was applied. Control refers to untreated chorionic villi and vehicle refers to DMSO treatment.

**Figure 2 antioxidants-10-00281-f002:**
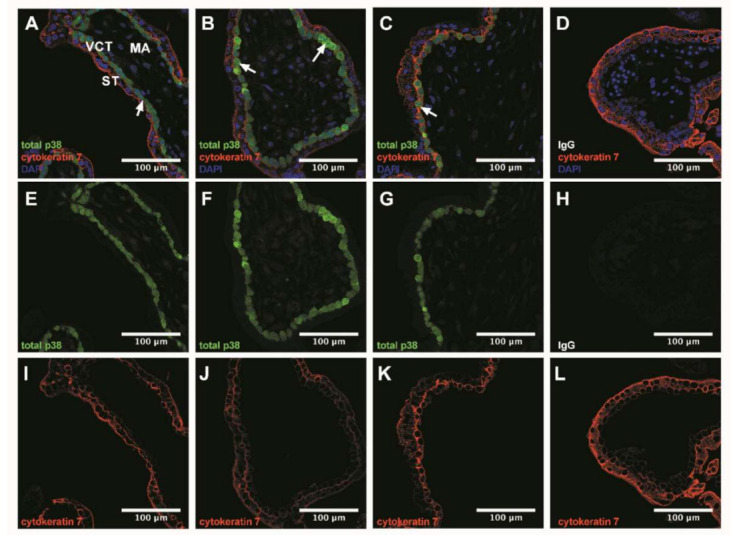
p38 immunodetection in 7–9 Gestational Weeks (GW) chorionic villi embedded in agarose sections. Total p38 was stained with green 488 AlexaFluor, nuclei were stained with DAPI and cytokeratin 7 with red 546 AlexaFluor. (**A**–**D**) Channels merge of total p38, cytokeratin 7 and DAPI staining; (**E**–**H**) Total p38 was shown in chorionic villi; (**I**–**L**) Structure of chorionic villi was marked with cytokeratin 7. IgG negative control is presented in (**D**,**H**,**L**). Confocal microscopy 400 x, scale bar 100 µm, *n* = 3 different placentas. ST, syncytiotrophoblast; VCT, villous cytotrophoblast; MA, mesenchymal axis.

**Figure 3 antioxidants-10-00281-f003:**
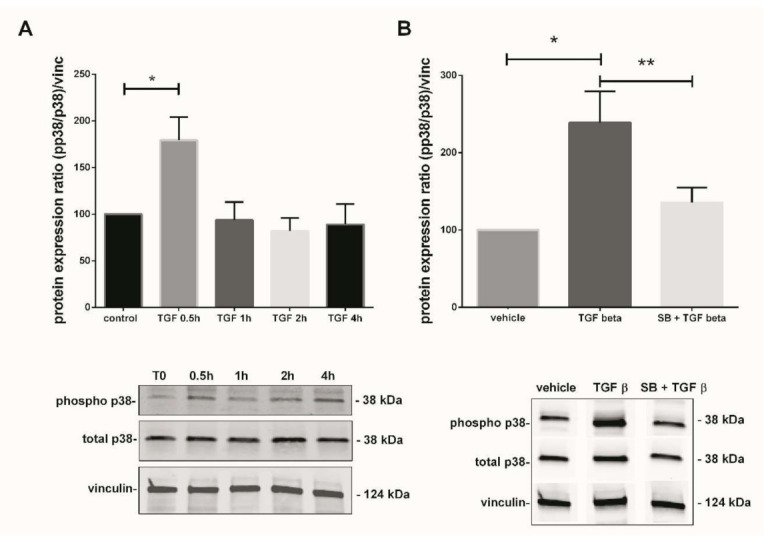
TGF-β1-mediated activation of p38 MAPK pathway in 7–9 Gestational Weeks (GW) chorionic villi. (**A**) Kinetic of p38 phosphorylation in first trimester chorionic villi exposed to 10 ng/mL of TGF-β1 for 0.5, 1, 2, or 4 h. Results are expressed as a variation of the ratio of phosphorylated to non-phosphorylated p38 protein expression as compared with time 0 (T0). The results are normalized to vinculin expression. Data are represented as mean +/− SEM, * *p* < 0.05, *n* = 7 different placentas. A one-way ANOVA test was applied. (**B**) Activation of p38 MAPK pathway in first trimester chorionic villi treated with TGF-β1 (10 ng/mL) for 48 h with or without pretreatment with SB203580 (10 µM) for 1 h. The protein expression of the phosphorylated p38/total p38 ratio was measured by Western blot. The ratio is significantly increased after 48 h of 10 ng/mL TGF-β1 treatment as compared with the untreated group and the p38 pretreatment with SB203580 inhibitor abolishes this effect. The protein expression was normalized to vinculin expression and reported to the expression of vehicle control. Data are represented as mean +/− SEM. * *p* < 0.05, ** *p* < 0.01, *n* = 11 different placentas. RM one-way ANOVA statistical analysis was applied. Control refers to untreated chorionic villi and vehicle refers to DMSO treatment.

**Figure 4 antioxidants-10-00281-f004:**
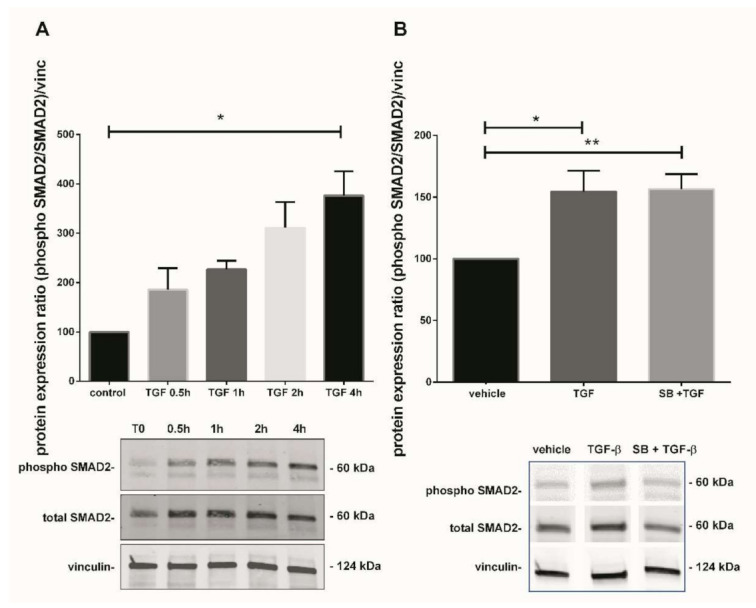
Effect of TGF-β1 on SMAD2 activation. (**A**) Kinetic of SMAD2 phosphorylation in 7–9 GW chorionic villi exposed to TGF-β1 at different times. The results are expressed as the phosphorylated SMAD2/total SMAD2 ratio and normalized on vinculin expression. Data are expressed as percentage of the control (T0) and represented as median +/− interquartile, *n* = 3 different placentas. * *p* < 0.05, Friedman test was applied; (**B**) Phosphorylation of SMAD2 protein in chorionic villi exposed to TGF-β1 for 48 h or SB203580 during 1 h prior to TGF-β1 treatment. Results are expressed as the phosphorylated SMAD2/total SMAD2 ratio and normalized on vinculin expression. Data are expressed as percentage of vehicle and represented as mean +/− SEM, *n* = 7 different placentas. * *p* < 0.05, ** *p* < 0.01, RM one-way ANOVA was applied. Control refers to untreated chorionic villi and vehicle refers to DMSO treatment.

**Figure 5 antioxidants-10-00281-f005:**
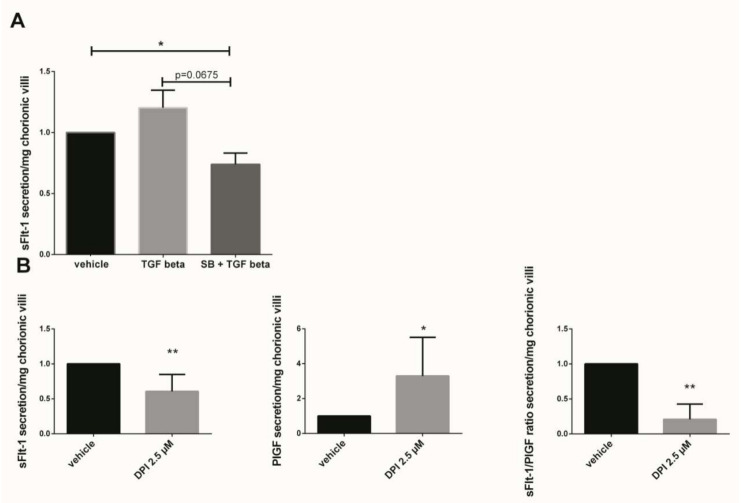
Effects of TGF-β1, p38 inhibitor (SB203580) and DPI on the sFlt-1 secretion by chorionic villi. (**A**) Chorionic villi were treated with TGF-β1 (10 ng/mL) for 48 h or pre-incubated with SB203580 (10 µM) for 1 h prior to the TGF-β1 treatment. The secretion of sFlt-1 was expressed as a change rate as compared with the vehicle group. Data are represented as mean ± SEM. Paired t-test were applied, *n* = 7 different placentas; (**B**) sFlt-1, PlGF secretion and sFlt-1/PlGF ratio in first trimester chorionic villi supernatant treated with 2.5 µM Nox inhibitor DPI. For sFLt-1, and PlGF secretion, data are represented as mean +/− SD, * *p* < 0.05, ** *p* < 0.01, *n* = 8 different placentas. Paired t-test was applied. For the sFlt-1/PlGF ratio, data are represented as mean +/− interquartile range. ** *p* < 0.01, *n* = 8 different placentas. Wilcoxon test was applied. Vehicle refers to DMSO treatment.

**Figure 6 antioxidants-10-00281-f006:**
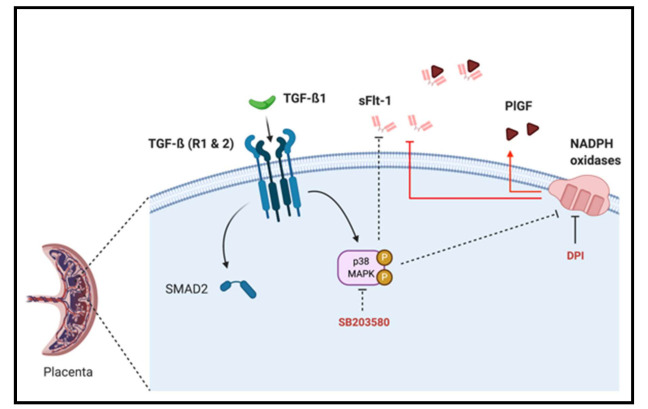
Schematic representation of TGF-β1/Nox/p38 MAPK mediated sFlt-1 secretion in first trimester chorionic villi. p38 MAPK can be activated by TGF-β1 and blockade of its activation decreases sFlt-1 secretion and placental Nox activity, and its activation is independent of TGF-β1 canonical pathway second messenger SMAD2. The modulation of the sFlt-1/PlGF angiogenic balance is under the control of Nox activity as its inhibition using DPI decreases sFlt-1 and increases PlGF secretions.

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
