# Peer review of "Human Placental NADPH Oxidase Mediates sFlt-1 and PlGF Secretion in Early Pregnancy: Exploration of the TGF-β1/p38 MAPK Pathways"

_antioxidants, 2021, doi:10.3390/antiox10020281_

Round 1
Reviewer 1 Report
This paper is potentially interesting but there are some issues that should be carefully addressed by authors before making the paper suitable for publication in the Antioxidants.
Specific comments and suggestions are given below.
Introduction is too long, a part of it can be moved to Discussion.
Lines 63-67: Please add references.
Lines 70-72: Please add references.
Lines 127-134: This section belongs to Conclusions.
Please add references in Methods where applicable.
Line 207: Please convert rpm to g.
Line 251: Please define Control and Vehicle in the text and in Figures.
Line 259: How did you choose tested concentrations?
Author Response
Dear Reviewer, please see the attachment.
Best regards,
Dr Zerrad-Saadi

Reviewer 2 Report
In this article the authors wanted to assess the possible signaling pathway of TGF-β1 / Nox / p38 induced sFlt-1 production in human chorionic villi (CV). The main focus of the manuscript was to investigate the complex relationship between the TGF-β1 and p38 MAPK pathways, the regulation of NADPH oxidase and the secretion of sFlt-1 and PIGF. They state that they have demonstrated for the first time that NADPH oxidase is involved in sFlt-1 and PIGF secretion in the first trimester chorionic villi. The authors highlight that the inhibition of Nox by DPI decreases sFlt-1 whilst PIGF secretions increase. They state that their data demonstrate the involvement of p38 MAPK in sFlt-1 secretion and Nox activation as blocking the p38 MAPK phosphorylation causes a decrease in both sFlt-1 secretion and also superoxide production.
The authors conclude that the placental NADPH oxidase play a major role in mediating the signal transduction cascade of sFlt-1 production. They suggest they demonstrate the involvement of p38 activation in the first trimester placental Nox activity.
Based on the data obtained by the authors, they suggest that their results highlight that Nox inhibition decreases the sFLT-1 / PIGF ratio. They do state however that they did not see any impact on TGF-β1-induced p38 MAPK activation on sFlt-1 secretion in the model they used. The authors conclude that their results emphasise a possible implication of the NADPH oxidases in early onset pathophysiology of preeclampsia through modulation of the angiogenic balance.
Main points and comments:
- MAPK is explained on line 102 having been cited with no explanation prior to that. Can this be corrected please?
- This article is basically well-written, reasonably easy to read and presents some interesting results. The tests employed all seem relevant and sensible for this type of research and the analysis is all very straightforward and appropriate for the study in question. Statistical analyses have been presented where needed.
- There are a few small issues with the English language or sentence structure (or an incorrect word) throughout the manuscript such as lines 38, 39, 53, 83, 90, 108, 114, 122, 176, 179, 183, 191, 195, 234, 264, 295, 300, 384, 392, 417, 504. I think the paper just needs a careful groom for minor inconsistencies that appear every now and then in order to make the final version more robust and easier to read.
- Line 131 has an incorrect word used. I think the authors may mean underlying as opposed to underlining?
- Can the authors please explain clearly what they mean by their statement in lines 179 – 180?
- Lines 216 - 217. Can the authors please be more specific with the concentration of the secondary antibody?
- Line 233. Can the authors explain their definition of “simplicate” in this sentence please?
- With regards to section 3.1, can the authors say whether there was a difference in the results obtained between the patients who were smokers and those who were not? Were the patient samples evenly distributed between the various assays used so that there were representative samples in each test for both smokers and non-smokers?
- Are the authors surprised they found no statistical significant difference between TGF-β1and SB+TGF-β1 treatment groups in Figure 1B?
- Figure 2 demonstrates some very clear confocal microscopy.
- Can the authors explain the lack of significance in Figure 3B between the vehicle and the SB+TGF-β1 groups? Did the authors expect the p38 pre-treatment with SB inhibitor to have such a marked effect?
- Where is Figure 4? It is referred to on line 280 and again on lines 330, 334 and 337. I can see a legend on lines 338 to 346 but no actual Figure. Please can the authors rectify this? Presumably this is just an oversight?
- In Figure 5A, were the authors surprised not to find statistically significant decreases in the sFlt-1 secretion in comparison with the TGF-β1 group following pre-incubation of the villi with p38 inhibitor SB203580?
- The authors conclude that their results demonstrate that the sFlt-1 and PlGF secretion in the first trimester chorionic villi are under regulation of Nox activity, and this results in the modulation of the angiogenic balance in first trimester chorionic villi. They observed a decrease of sFlt-1 secretion and an increase of PlGF secretion in the presence of the Nox inhibitor DPI and the authors say this supports the relevance of their initial hypothesis as stated on lines 382 to 385. The authors also suggest that their results highlight a crosstalk between the Nox and p38 MAPK pathway although the mechanism of action remains unclear.
- Line 398. Can the authors suggest any modifications to their current model and experimental design that may give rise to more robust data analysis and allow statistically significant differences to be uncovered?
- Lines 399 and 400 suggest that the expression of both catalytic and cytosolic subunits of the Nox isoform remains to be tested as the experimental design used for the present manuscript is unable to demonstrate a significant effect of TGF-β1 on Nox activity. Are the authors satisfied that they are presenting robust data in this manuscript or is further analysis and data collection required? There are still many unanswered questions and this article appears to have raised a few more questions. I appreciate that this is the case with most scientific analyses.
- Lines 410 to 412 highlight the possibility that the authors may have missed the window of opportunity to see superoxide anion production by the chorionic villi. Further experimentation is required in order to clarify and confirm whether this is the case. As the authors state themselves, the lack of data demonstrating a link between the TGF-β1 pathway, p38 MAPK activation and sFlt-1 secretion in the first trimester chorionic villi may well indicate that other pathways are involved and the system is far more complex than first anticipated.
- The authors make a very valid point that TGF-β1 does seem to have a variety of effects depending on the cell type. They also make the point that sFlt-1 secretion may be under the control of a number of independent activator pathways, including TGF-β1-mediated p38 MAPK activation and NADPH oxidase ROS production in the first trimester chorionic villi. Figure 6 highlights that p38 MAPK phosphorylation seems to be independent of SMAD2 activation. Based on the data shown and the in depth discussion, the authors do mention the pros and cons of the research area under scrutiny and they do highlight the assessments that are still in need of further analysis. Their results have generated novel findings and the authors are well aware of the limitations of their own study and sample sets. Their conclusions are reasonably well balanced and fair.
- The Supplementary Figures are not mentioned in the text (unless I have completely missed reference to them). Can they please be included? Each of the Supplementary files requires a Figure number as they are all labelled as “Supplementary data” at the moment. It is actually a shame that these data cannot be shown in the main body of the article rather than as Supplementary data sets.
Author Response

(The authors gave the same response as above.)

Round 2
Reviewer 1 Report
Authors significantly improved the manuscript.